# Psychological Interventions in a Pandemic Emergency: A Systematic Review and Meta-Analysis of SARS-CoV-2 Studies

**DOI:** 10.3390/jcm11113209

**Published:** 2022-06-04

**Authors:** Grazia D’Onofrio, Nicoletta Trotta, Melania Severo, Salvatore Iuso, Filomena Ciccone, Anna Maria Prencipe, Seyed Mohammad Nabavi, Gabriella De Vincentis, Annamaria Petito

**Affiliations:** 1Clinical Psychology Service, Health Department, Fondazione IRCCS Casa Sollievo della Sofferenza, San Giovanni Rotondo, 71013 Foggia, Italy; f.ciccone@operapadrepio.it (F.C.); am.prencipe@operapadrepio.it (A.M.P.); 2Department of Clinical and Experimental Medicine, University of Foggia, 71122 Foggia, Italy; n.trotta2@gmail.com (N.T.); melania.severo93@gmail.com (M.S.); iuso.salvatore@libero.it (S.I.); annamaria.petito@unifg.it (A.P.); 3Advanced Medical Pharma (BIOTEC), 82100 Benevento, Italy; nabavi208@gmail.com; 4Health Department, Fondazione IRCCS Casa Sollievo della Sofferenza, San Giovanni Rotondo, 71013 Foggia, Italy; g.devincentis@operapadrepio.it

**Keywords:** viral infection, psychological treatment, depression, anxiety, resilience

## Abstract

Background: The study aim was to review the evidence and effectiveness of psychological interventions applied during Severe Acute Respiratory Syndrome Coronavirus 2 (SARS-CoV-2) pandemic. Methods: A literature search was run from April 2020 to April 2021. The inclusion criteria were: (1) RCTs or observational studies; (2) professional health carers and patients who had contracted coronavirus during the pandemic; (3) adults and elderly people with a viral infection diagnosis; (4) suitable measures to assess intervention effectiveness on clinical status and psychological and behavioral aspects. An internal validity assessment was planned using robvis. Data were synthesized according to PICO criteria. Results: A total of 12 studies were selected. Studies measuring mental health outcomes demonstrated the greatest reduction in symptoms, with eight out of the twelve studies demonstrating a reduction in symptoms that reached statistical significance (*p* < 0.05) and four of the studies reaching a higher significance level of *p* < 0.01. The type of psychological intervention was predominantly cognitive behavioral therapy (CBT). All studies except one was run online. Conclusion: Psychological interventions have a benefit on mental health outcomes, even if performed online. In particular, CBT seems to be the psychological intervention that is used more and also seems to have a larger effect size on the mitigation of mental health symptoms and increasing resilience.

## 1. Introduction

In the past 10 years, several viral epidemics have occurred, such as influenza caused by the virus subtype H1N1 (2009), Middle East respiratory syndrome—MERS (2012), Ebola virus disease (2014) [1,2,3], and a novel virus Severe Acute Respiratory Syndrome Coronavirus 2 (SARS-CoV-2) that emerged around the world at the end of 2019. Globally, response measures were taken, including self-isolation, social distancing, and quarantines, augmenting further the distress.

According to several current studies about the prevalence of psychological impacts resulting from SARS-CoV-2, healthcare workers and infected/recovered individuals were those who suffered most from psychiatric morbidity, post-traumatic stress disorders (PTSD), anxiety, and depression (respectively, healthcare workers: 21–29%; infected/recovered individuals: 25–56%), followed by adults in the community (15–19%), and quarantined adults (12%) [4,5,6,7].

Compared to previous pandemic (H1N1, MERS, and Ebola) studies, psychological impact prevalence estimated during SARS-CoV-2 were higher [5]. In particular, across all populations psychiatric morbidity was the most prevalent disorder (32%), followed by PTSD (21%), depression (17%), and anxiety (12%).

During the SARS-CoV-2 pandemic, innovative technological know-how displayed a great ability to back up public health interventions [8], permitting an effective psychological assistance for patients during the pandemic with a positive impact on mental health aspects. However, evidence-based advice on what would be effective in mitigating this impact is not yet clear. A rapid review was conducted to detect the most effective interventions for reducing nurses’ psychological issues during outbreaks [9], but the 14 articles included were the recommendations of organizational and individual self-care interventions, except for a study which described a digital intervention and user satisfaction. Another rapid systematic review was conducted to identify coping strategies among healthcare workers [10], but it only included studies with at least 100 healthcare workers, and removed studies with smaller samples that might contribute to clinical practice and future research.

Therefore, the aim of this work was to review the evidence and effectiveness of psychological interventions applied during the SARS-COV-2 pandemic, including therapeutic outcomes and measurable psychological improvements (reduction in depression and/or anxiety and/or post-traumatic stress symptoms from baseline to the last available follow-up, evaluated using suitable measures to assess psychological and behavioral aspects and clinical status).

## 2. Materials and Methods

### 2.1. Search Strategy

Prospero pre-registration was run with n. CRD42020219191 (9 November 2020), available at: https://www.crd.york.ac.uk/prospero/display_record.php?ID=CRD42020219191 (accessed on 12 April 2022).

A literature search was run using Scopus, MEDLINE (Web of Science), CINAHL Complete, PsycINFO, and PubMed (from April 2020 to April 2021).

The PRISMA guidelines for systematic reviews were followed [11] and employed to evaluate the quality of the study. An inductive approach was applied to the analysis [12].

The search queries included terms (Severe Acute Respiratory Syndrome Coronavirus 2 OR Coronavirus OR SARS-CoV-2 OR COVID OR COVID-19) to determine the interest outcomes (viral infection AND [psychological treatment OR psychological first aid OR psychological support OR psychotherapy sessions]).

The inclusion criteria are shown below: (1) RCTs or observational studies; (2) professional health carers and patients who contracted coronavirus during the pandemic; (3) adults and elderly people with viral infection diagnosis; (4) suitable measures to assess intervention effectiveness on clinical status and psychological and behavioral aspects. The exclusion criteria were the following: (1) participants with severe mental health symptoms such as psychosis or suicidality; (2) studies that were not published in English, and (3) no psychological interventions.

### 2.2. Study Selection

Three reviewers applied eligibility criteria and selected studies for inclusion in the systematic review. Of them, two reviewers screened records for inclusion and one reviewer checked decisions in a blinded manner.

The characteristics to assess were the following: randomization, deviations from the intended intervention, missing data, outcome measurement, and selection of reported results.

The assessment was conducted at study and outcome level.

The criteria used to assess the internal validity was planned using robvis (a visualization tool for risk of bias assessments in a systematic review) [13]. Inter-rater reliability was measured using Cohen’s kappa, k with −1 (absence of an agreement) to 1 (perfect agreement) [14].

Three researchers extracted and checked the received data. Of them, two researchers independently extracted data and one researcher checked the extracted data and quality assessment.

### 2.3. Data Synthesis

The recorded data were inserted into one .xls spreadsheet and analyzed using R Ver. 2.8.1 statistical software package (The R Project for Statistical Computing; available at URL http://www.r-project.org/, accessed on 11 April 2022). The data to be extracted from the studies were the following: sample size, sample type (patients, clinicians, nurses, other health professionals), mean age ± sd, psychological intervention type, performed session number, session modality (onsite or online), performed measurements, and achieved outcomes (pre and post intervention). The main outcome(s) was reduction in depression and/or anxiety and/or post-traumatic stress symptoms from baseline to the last available follow-up, evaluated using suitable measures to assess psychological and behavioral aspects and clinical status.

The data were synthesized according to PICO criteria [15], consisting of the testing for a relationship between the size and precision of study effects, the risk ratios for individual studies combined using a random effects meta-analysis, and co-authors supplying a complete summary and accurate organization of all studies covering transparency, approach, methodology, and strengths/weaknesses [16,17].

## 3. Results

The details of the literature search and study selection procedure are outlined in the PRISMA flow diagram (Figure 1).

As shown in Figure 1, a total of 1416 articles were identified, of which 103 duplicates were deleted. After abstract evaluation, 1169 studies were excluded. Another 132 studies were excluded after a more comprehensive examination. Consequently, 12 published studies were suitable for this systematic review.

### 3.1. Methodological Quality

According to Cochrane’s recommendations, the robvis tool was used in order to assess bias risk (Figure 2), rating bias as “low”, “high”, or “some concerns” against several domains.

The nature of the interventions meant that most studies presented a trend towards a low risk of bias. At this point in the review, a meta-analysis was considered appropriate.

### 3.2. Data Extraction and Outcomes

A data extraction table was created in order to standardize the process (Table 1). Contacting the authors was unnecessary because no missing information was identified.

### 3.3. General Overview of Included Studies

Twelve studies of psychological interventions on people during the SARS-CoV-2 pandemic were identified for inclusion [18,19,20,21,22,23,24,25,26,27,28,29]. All studies were published in 2020 and 2021 and published in indexed journals.

All studies, except for Shaygan et al. [22], occurred in countries of high economic development.

The study aims were to treat mental health symptoms, with most participants having baseline symptoms classed as clinical disease.

The sample sizes ranged from 9 participants to 1500 participants. Seven studies included patients, of which three studies included people with a mean age ranging from 40.8 to 48.3 years [19,24,25], two studies included people with a mean age ranging from 34.6 to 36.8 years [22,26], and two studies included younger people with a mean age ranging from 26.1 to 28.5 years [23,28]. Another study included health care workers without personal identifying information being collected [18]. The last four studies included the general population/adults with a mean age ranging from 42.0 to 45.0 years [20,29] and students with a mean age ranging from 8.2 to 31.0 years [21,27].

All studies, except for Mellins et al. [18], Wei et al. [19], and Schlarb et al. [29], were randomized controlled trials (RCT). In particular, one study was a cluster-randomized controlled trial (cRCT) [22], one study was an open-label pragmatic randomized controlled trial (opRCT) [23], one study was a prospective multicenter two-arm randomized controlled trial (PM2aRCT) [25], and one study was a longitudinal two-arm clinical trial (L2aCT) [26].

Finally, the three no-RCT studies were a validation protocol study [18], a case–control study [19], and a pilot study [29].

The type of psychological intervention was predominantly cognitive behavioral therapy (CBT) [18,19,20,21,23,24,25,26,27,28,29], and the total duration of therapy ranged from 10 min to 1 h per day [19,22,25] and per week [21,23,24,26,29]. Of them, in two studies, the psychological intervention type was mindfulness-based intervention (MBI) [21,28]; Malboeuf-Hurtubise et al. [21] compared MBI with a psycho-educational approach with a session duration of 1 h per week. Three studies did not clearly specify the number of sessions performed and session duration [18,20,27].

Only in two studies was the type of psychological intervention a psycho-educational approach [21,22] with a session duration of 1 h per week [21] and per day [22].

All studies were run online.

The studies had different durations: one study lasted 3 months [18], and eleven studies ranged from 1 to 7 weeks [19,20,21,22,23,24,25,26,27,28,29]. The follow-up period (1 month) was taken into account only in three studies [20,24,25].

### 3.4. Outcomes

Post-intervention, the mental health outcome scores were significantly improved compared to the pre-intervention scores for all twelve studies (Table 2). The Cohen’s d effect sizes (where calculated) were small to large (d = 0.21–2.89), indicating an increase in primary/secondary coping abilities [27,28], and a decrease in depression and anxiety [20,26,28], functional impairment [20], intolerance of uncertainty [20], and insomnia [20,26,29]. However, in two other studies the d_ppc2_ and d*f* effect sizes were calculated showing, respectively, large and medium-large effect sizes in resilience increase and perceived stress reduction [22], and a very large effect size in depression and anxiety reduction [23].

The partial eta squared (ηp 2) effect size was calculated in three studies, revealing a significant large effect (ηp 2 = 0.20–0.89) on mental health condition with a decrease in depression and anxiety symptoms [21,24,25], functional and attention deficits [21], and traumatic symptoms [24]. A medium effect (ηp 2 = 0.07–0.08) was reported in insomnia reduction [25].

A study reported that post hoc analyses of the individual time points showed that depression and anxiety were significantly decreased (Depression: F = 37.35, *p* < 0.001; Anxiety: F = 26.58, *p* < 0.001), as well as a main effect of the group (Depression: F = 4.384, *p* = 0.047; Anxiety: F = 5.634, *p* = 0.026) and a group-by-time interaction (Depression: F = 5.268, *p* = 0.009; Anxiety: F = 3.743, *p* = 0.031) [19].

Only one study did not run the effect size calculation, showing a significant reduction in emotional distress (pre-intervention: 2.6 ± 0.9; post-intervention: 2.0 + 0.8, *p* = 0.03) [18].

More specifically, the studies measuring mental health outcomes demonstrated the greatest reduction in symptoms (Figure 3), with eight out of the twelve studies demonstrating a reduction in symptoms that reached statistical significance (*p* < 0.05) [18,22,23,25,26,27,28,29], and four of the studies reaching a higher significance level of *p* < 0.01 [19,20,21,24].

There were no instances of the intervention group having worse scores post intervention. In contrast, in three of the eight studies with control groups the mental health scores worsened over the study period [22,23,24]. Only two study with one-month follow up period, post-intervention outcome scores were maintained [20,25].

## 4. Discussion

According to the study aim, the evidence and effectiveness of psychological interventions applied during the SARS-CoV-2 pandemic, including therapeutic outcomes and measurable psychological improvements, were reviewed.

In the meta-analysis, it showed that psychological interventions have a benefit on mental health outcomes, even if performed online. In particular, CBT seems to be the psychological intervention that is used more and seems to have a larger effect size on the mitigation of mental health symptoms and increasing resilience.

CBT has a positive impact on symptoms such as depression, anxiety, and perceived stress in health care workers, patients, general population, and students. The American Psychological Association and International Society of Traumatic Stress Studies specified to constantly monitor the mental health status of pandemic-affected populations and provide timely evidence-based trauma-centered psychotherapies, such as CBT [30,31,32].

However, beyond the above, the importance of enhancing the online psychological support has emerged, because of traditional face-to-face psychological facilities being unavailable during pandemic outbreak [32]. Tele-medical approaches needed to be implemented in public health strategies in order to support many people simultaneously and effectively [33]. The benefits of e-mental health approaches are great, and these innovative resources are currently more necessary than ever before [34].

Given the evidence for the risk of short- and long-term psychological consequences and their impact on job performance and quality of care [35], psychological packages should be urgently implemented to manage mental health care for medical staff working on the frontline against the pandemic [36].

The studies by Brooks et al. [37], in the context of the psychological effects on people due to other epidemics (Ebola and SARS), identify five main causes of stress related to lockdown: (1) duration of the lockdown; (2) fear of infection; (3) feelings of frustration and boredom; (4) inadequate devices or unobtainable masks; (5) inadequate information or an excessive increase in information sources.

To these factors, we can also add concerns related to the financial situation that have generated negative feelings in some individuals and symptoms attributable to psycho-emotional stress.

Furthermore, it is necessary to identify how certain characteristics inherent in factors such as demand, control, and support play a role in modulating the perception of stress [38]. Therefore, it is necessary to bring the reflection on stress closer to the coping constructs, highlighting how individual motivation to a task influences the perception of the situation, thus modulating the level of stress. The contribution of Caplan and Van Harrison relates to the observation that, alongside stress, adaptation coexists as a possible adaptive response; this outcome is the result of an interaction between person and environment to which objective and subjective variables present both in the person and in the environment contribute. Regarding the included studies in our work, we found limited evidence for coping capacity. Predominantly, the studies have a low risk of bias, but we only dealt with the main effects. Moreover, not all the variables were comparable, in particular the heterogeneity of size effect assessment. It is necessary for future studies to improve design quality to increase the trust in their outcomes.

Therefore, psychological systems as well as medical systems of health care require huge investment and major upgrading in order to be prepared for future pandemics.

## 5. Conclusions

An excellent way to improve mental health is to focus on reducing social inequity through an integrated community-oriented system of care operating across health and social care systems, supported by operational research to guide implementers and policymakers to deal with current and future challenges. This requires political will at the highest policy-making level, and an increase in resources at all levels of health and social care systems [39,40].

## Figures and Tables

**Figure 1 jcm-11-03209-f001:**
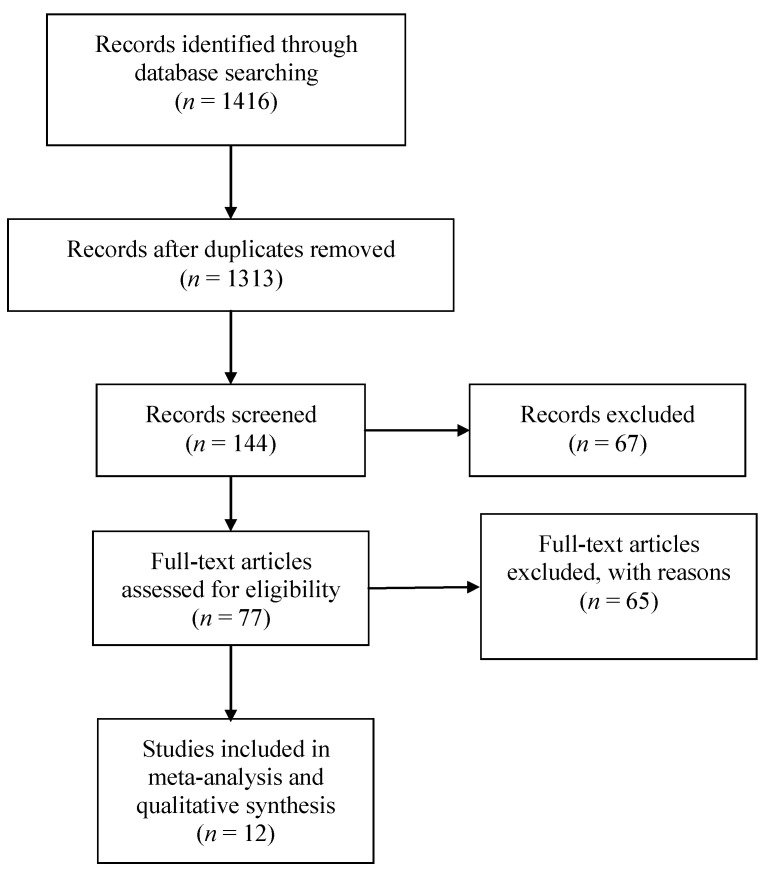
Flow diagram outlining the selection procedure to identify articles which were included in the analysis of psychological interventions in SARS-CoV-2 pandemic emergency.

**Figure 2 jcm-11-03209-f002:**
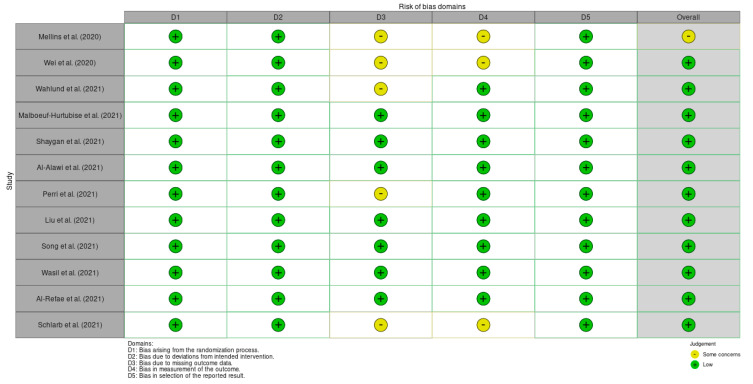
Risk of bias in included studies.

**Figure 3 jcm-11-03209-f003:**
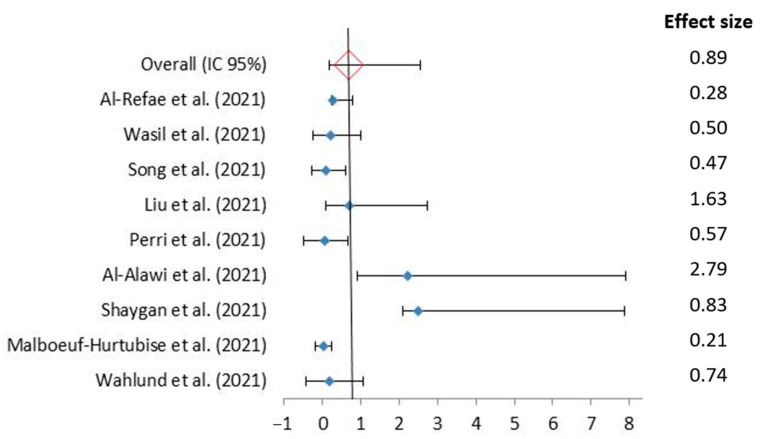
Forest plot of the psychological intervention effect during the SARS-CoV-2 pandemic Using Random Effects Model [20,21,22,23,24,25,26,27,28]. Squares indicate the OR of individual studies, and the extended lines denote 95% confidence intervals (CI). The diamond data indicates pooled prevalence. Test of heterogeneity: I^2^ = 99.99%, *p* = 0.00.

**Table 1 jcm-11-03209-t001:** A general overview of the included studies.

Study	Pandemic	Country	Study Design	Sample Size	Sample Type	Age Mean ± SD	Psychological Intervention Type	Number of Sessions Performed	Session Duration	Session Modality	Duration	Ref.
Mellins et al. (2020)	SARS-CoV-2	USA	VPS	186 groups of 1–30 people (124 completed the survey)	Health care workers	ns	CBT ACT	30 groups met once 22 groups met 2–4 times 9 groups met ≥5 times	30 min.	online	3 months	[18]
Wei et al. (2020)	SARS-CoV-2	China	CCS	26 participants	Patients	40.8 ± 13.5	ICBT	1/day	50 min.	online	2 weeks	[19]
Wahlund et al. (2021)	SARS-CoV-2	Sweden	RCT	335 participants 335 controls	General population	45.0 ± 13.0 47.0 ± 14.0	CBT	5 modules	ns	online	3 weeks + 1 month follow-up	[20]
Malboeuf-Hurtubise et al. (2021)	SARS-CoV-2	Canada	RCT	37 children	Students	8.2 ± ns	P4C MBI	1/week	1 h	online	5 weeks	[21]
Shaygan et al. (2021)	SARS-CoV-2	Iran	cRPC	26 participants 22 controls	Patients	36.8 ± 11.8	OMPI	1/day	1 h	online	2 weeks	[22]
Al-Alawi et al. (2021)	SARS-CoV-2	Oman	opRCT	22 participants 24 controls	Patients	28.5 ± 8.7	CBT ACT	1/week	1 h	online	6 weeks	[23]
Perri et al. (2021)	SARS-CoV-2	Italy	RCT	38 participants	Patients	48.3 ± 13.6 52.4 ± 10.6	EMDR TF-CBT	2/week	ns	online	3 weeks + 1 month follow-up	[24]
Liu et al. (2021)	SARS-CoV-2	China	PM2aRCT	126 participants 126 controls	Patients	43.8 ± 14.3 41.5 ± 11.5	cCBT + TAU TAU (control group)	1/day	10 min.	online	1 week + 1 month follow-up	[25]
Song et al. (2021)	SARS-CoV-2	China	L2aCT	83 participants 46 controls	Patients	34.6 ± 9.1	MiCBT	1/week	30 min.	online	3 weeks	[26]
Wasil et al. (2021)	SARS-CoV-2	USA	RCT	189 participants 24 controls	Students	31.0 ± 8.9	COMET (based on CBT + gratitude intervention)	Single session	20–25 min.	online	ns	[27]
Al-Refae et al. (2021)	SARS-CoV-2	Canada	RCT	78 participants 87 controls	Patients	26.1 ± 8.4 24.5 ± 8.9	MBI SERENE	At least 1/day	ns	online	4 weeks	[28]
Schlarb et al. (2021)	SARS-CoV-2	Germany	Pilot study	9 participants	Adults	42.0 ± ns	iCBT-I	1/week	8.34 ± 2.32	online	7 weeks	[29]

Legend: VPS, validation protocol study; CCS, case–control study; RCT, randomized controlled trial; cRPC, cluster randomized parallel controlled; opRCT, Open-label, Pragmatic, Randomized Controlled Trial; PM2aRCT, prospective multicenter two-arm randomized controlled trial; L2aCT, longitudinal two-arm clinical trial; ns, not specified; CBT, cognitive behavioral therapy; ACT, acceptance and commitment therapy; ICBT, internet-delivered cognitive behavior therapy; MBI, mindfulness based interventions; P4C, philosophy for children; OMPI, online multimedia psychoeducational intervention; EMDR, eye movement desensitization and reprocessing; TF-CBT, trauma-focused cognitive behavioral therapy; cCBT, computerized cognitive behavioral therapy; TAU, treatment as usual; MiCBT, mobile internet cognitive behavioral therapy; COMET, Common Elements Toolbox; SERENE, self-compassion-based cognitive smartphone intervention; iCBT-I, Internet-based cognitive behavioral therapy for insomnia.

**Table 2 jcm-11-03209-t002:** Visual display of outcomes table.

Study	Ref.	PTSD	Depression	Anxiety	Primary/Secondary Coping	Functional Impairment	Intolerance of Uncertainty	Insomnia	Attention Deficit	Resilience	Perceived Stress	Performed Measures
Mellins et al. (2020)	[18]	—	—	—	—	—	—	—	—	—	↓ *	Survey Qualtrics
Wei et al. (2020)	[19]	—	↓ **	↓ **	—	—	—	—	—	—	—	17-HAMD HAMA
Wahlund et al. (2021)	[20]	—	↓ **	↓ **	—	↓ **	↓ **	↓ **	—	—	—	GAD-7 MADRS-S WSAS IUS-12 ISI
Malboeuf-Hurtubise et al. (2021)	[21]	—	↓ **	↓ **	—	↓ **	—	—	↓ **	—	—	BASC III BPN
Shaygan et al. (2021)	[22]	—	—	—	—	—	—	—	—	↑ *	↓ *	CD-RISC PSS CSQ-I
Al-Alawi et al. (2021)	[23]	—	↓ *	↓ *	—	—	—	—	—	—	—	PHQ-9 GAD-7
Perri et al. (2021)	[24]	↓ **	↓ **	↓ **	—	—	—	—	—	—	—	PCL-5 STAI-Y1 BDI-II
Liu et al. (2021)	[25]	—	↓ *	↓ *	—	—	—	↓ *	—	—	—	17-HAMD HAMA SDS SAS AIS
Song et al. (2021)	[26]	—	↓ *	↓ *	—	—	—	↓ *	—	—	—	PHQ-9 GAD-7 ISI CD-RISC VAS
Wasil et al. (2021)	[27]	—	—	—	↑ *	—	—	—	—	—	—	SCSC adapted 3-items
Al-Refae et al. (2021)	[28]	—	↓ *	↓ *	↑ *	—	—	—	—	—	↓ *	DASS-21 SD-WISE PWBS
Schlarb et al. (2021)	[29]	—	—	—	—	—	—	↓ *	—	—	—	PSQI

Legend: The arrows denote the direction of change in the mental health outcome measures. * = statistically significant result, whereby *p* ≤ 0.05; ** = statistically significant result, whereby *p* ≤ 0.01. 17-HAMD, 17-item Hamilton Depression Scale; HAMA, Hamilton Anxiety Scale; GAD-7, Generalized Anxiety Disorder seven item scale; MADRS-S, Montgomery Åsberg Depression Rating Scale – Self report; WSAS, Work and Social Adjustment Scale; IUS-12, Intolerance of Uncertainty Scale short version; ISI, Insomnia Severity Index; BASC III, Behavior Assessment Scale for Children-3rd edition, self-report scale; BPN, basic psychological need satisfaction; CD-RISC, Connor-Davidson resilience scale; PSS, Perceived Stress Scale; CSQ-I, client satisfaction questionnaire adapted to internet-based interventions; PHQ-9, Patient Health Questionnaire-9; PCL-5, post-traumatic stress disorder checklist for DSM 5; STAI-Y1, State Trait Anxiety Inventory; BDI-II, Beck Depression Inventory-II; SDS, Self-rating Depression Scale; SAS, Self-rating Anxiety Scale; AIS, Athens Insomnia Scale; VAS, visual analogue scale (anxiety self-rating scale of COVID-19); SCSC, Secondary Control Scale for Children; DASS-21, Depression, Anxiety, and Stress Scale; SD-WISE, San-Diego Wisdom scale; PWBS, Psychological Well-being scale; PSQI, Pittsburgh Sleep Quality Index.

## Data Availability

Not applicable.

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
