# Peer review of "Psychological Interventions in a Pandemic Emergency: A Systematic Review and Meta-Analysis of SARS-CoV-2 Studies"

_jcm, 2022, doi:10.3390/jcm11113209_

Round 1
Reviewer 1 Report
Interesting article on the analysis of the use of different forms of psychotherapy during a pandemic.
The article is correctly written and an adequate methodology has been used.
It concludes with interesting results that psychological interventions affect resilience.
Author Response
- Interesting article on the analysis of the use of different forms of psychotherapy during a pandemic. The article is correctly written and an adequate methodology has been used. It concludes with interesting results that psychological interventions affect resilience. 1. We thank the reviewer for his/her positive feedback.
Reviewer 2 Report
Thanks for opportunity to review manuscript entitled ‘‘ Psychological Interventions in a Pandemic Emergency: Systematic Review and Meta-analysis of 10-year studies’’. This article examined used interventions in pandemic times and effectiveness of these interventions. Overall, as a multivariate analyst and experienced reviewer, I think this article is very problematic from beginning to end that prevent me positive evaluation of manuscript. I summarized very briefly some of them as below.
- Author reviewed the last ten year studies in their meta-analyses. The reason and purpose of this did not given in Introduction section.
- In the Introduction section of manuscript, each paragraph consists of only one or two sentences. As a general recommendation, each paragraph must consist of three to eight sentences. Moreover, Introduction section is very inadequate for a scientific article. Authors did not give any information about previous systematic reviews and meta-analyses and weakness of these studies. Moreover, what their study add to existing literature also unclear. Why specifically they choose to work last ten years also missing in Introduction section. Overall, Introduction section give nothing readers about importance of this study.
- Method section must reconstruct using subtitles for better flow. The same problem in the Introduction section also exist in this section. each paragraph consists of only one or two sentences. Moreover, I think some sentences are not in English language rather Italian languages such as The Materials Prospero pre-registration n. CRD42020219191(November 9, 2020) available at: https://www.crd.york.ac.uk/prospero/display_record.php?ID=CRD42020219191. 61
- Authors used three different database to identify potential studies. However, they did not used most trustable data base for systematic review and meta-analysis namely Web of science.
- What authors want to mean with no English editing is unclear.
- Hoe authors identified the last ten year pandemic is unclear. Which references used to identify pandemics in last ten yea in keywords is unclear.
- Interrater reliability of coding procedure must be added to Method section.
- Which method authors used in meta-analyses is unclear random effect of fixed effect meta-analyses and reasons behind using them.
- Why authors did not conduct a moderator analyses are unclear.
- Why authors did not used effect size conversation formula add report all studies using Cohen d is unclear.
- What authors want to mean with two studies dppc2 and df effect sizes were calculated. I did not know such effect sizes as a multivariate analyst.
- I am not able o find meta-analysis results in results section. Authors reported findings from literature review.
- Discussion section must completely rewritten. It is not based on study findings. Moreover, limitation section and practical implications section completely missing in the manuscript.
- Overall, I think author must rewrite this article again from beginning to end and should solely focus on Covid-19 pandemic. If table 1 correct, it is most logical for them.
Author Response
- Thanks for opportunity to review manuscript entitled ‘‘ Psychological Interventions in a Pandemic Emergency: Systematic Review and Meta-analysis of 10-year studies’’. This article examined used interventions in pandemic times and effectiveness of these interventions. Overall, as a multivariate analyst and experienced reviewer, I think this article is very problematic from beginning to end that prevent me positive evaluation of manuscript. I summarized very briefly some of them as below. We hope for reviewer reconsideration about our work after revision of the document according his/her constructive observations. Below, please find item-by-item responses to your comments, which are included verbatim.
- Author reviewed the last ten year studies in their meta-analyses. The reason and purpose of this did not given in Introduction section. We re-wrote the article from beginning to end solely focusing on Covid-19 pandemic. We clarified the study purpose, as reviewer observed.
- In the Introduction section of manuscript, each paragraph consists of only one or two sentences. As a general recommendation, each paragraph must consist of three to eight sentences. We organized the paragraphs according to reviewer suggestion.
- Moreover, Introduction section is very inadequate for a scientific article. Authors did not give any information about previous systematic reviews and meta-analyses and weakness of these studies. Moreover, what their study add to existing literature also unclear. We added the requested information about previous systematic reviews, according to reviewer suggestion.
- Why specifically they choose to work last ten years also missing in Introduction section. Overall, Introduction section give nothing readers about importance of this study. According to reviewer suggestions (in particular to point 17), we rewrote the article from beginning to end solely focusing on Covid-19 pandemic.
- Method section must reconstruct using subtitles for better flow. The same problem in the Introduction section also exist in this section. each paragraph consists of only one or two sentences. We re-organized the Method section, according to reviewer suggestions.
- Moreover, I think some sentences are not in English language rather Italian languages such as The Materials Prospero pre-registration n. CRD42020219191(November 9, 2020) available at: https://www.crd.york.ac.uk/prospero/display_record.php?ID=CRD42020219191. 61 According to reviewer suggestion, we obtained a review of the manuscript from a native English speaker.
- Authors used three different databases to identify potential studies. However, they did not used most trustable data base for systematic review and meta-analysis namely Web of science. We used Web of Science too. According to reviewer observation, we added this information in the text.
- What authors want to mean with no English editing is unclear. We correct the sentence in the following form: “studies did not publish in English”.
- How authors identified the last ten-year pandemic is unclear. Which references used to identify pandemics in last ten year in keywords is unclear. We deleted the studies previous to Covid pandemic.
- Inter rater reliability of coding procedure must be added to Method section. Which method authors used in meta-analyses is unclear random effect of fixed effect meta-analyses and reasons behind using them.
We added the missing information, according to reviewer comments.
- Why authors did not conduct a moderator analyses are unclear. We could not perform the moderator analysis because we did not have the databases used in the studies included in this meta-analysis.
- Why authors did not used effect size conversation formula add report all studies using Cohen d is unclear. We reported the effect sizes as calculated by the authors of the studies included in the meta-analysis. We therefore decided not to modify them with the aim of making the reported data more easily identifiable.
- What authors want to mean with two studies dppc2 and df effect sizes were calculated. I did not know such effect sizes as a multivariate analyst. As above, we reported the effect sizes as calculated by the authors of the studies included in the meta-analysis.
- I am not able o find meta-analysis results in results section. Authors reported findings from literature review. We added a Forest plot (Figure 3) in order to clarify the meta-analysis outcomes.
- Discussion section must completely rewritten. It is not based on study findings. Moreover, limitation section and practical implications section completely missing in the manuscript. We rewrote the Discussion section, according to reviewer observations.
- Overall, I think author must rewrite this article again from beginning to end and should solely focus on Covid-19 pandemic. If table 1 correct, it is most logical for them. We rewrote the article from beginning to end solely focusing on Covid-19 pandemic. We thank the reviewer again for his/her suggestions that have certainly improved and enriched our work.
Reviewer 3 Report
Dear authors,
Thank you for allowing me to review this important work. I have a few suggestions/queries :
1) In the methodology section, the authors do not mention which databases the studies were extracted from. Please include this information.
2) The inclusion and exclusion criteria for the selected studies are not included.
3) There is a stark absence of a forest plot to summarize the findings of the meta-analysis. Please include one if possible.
4) The discussion section is rather simplistic and brief. It should be more scholarly in nature and discuss the novelty of the findings in a critical way.
5) There are many serious grammatical errors throughout the manuscript. The authors are advised to thoroughly proofread the manuscript before resubmission.
6) Important limitations and strengths of the meta-analysis must be mentioned in the manuscript.
6) In the introduction section, lines 36-40, the authors posit that the pandemic has lead to many psychological consequences. The authors are guided to the following papers that also elaborate on the mental health outcomes of frontline workers during the pandemic :
1) Francis B, Soon Ken C, Yit Han N, et al. Religious coping during the COVID-19 pandemic: gender, occupational and socio-economic perspectives among Malaysian frontline healthcare workers. Alpha Psychiatry. 2021;22(4):194-199.
2) Narendra Kumar, M. K., Francis, B., Hashim, A. H., Zainal, N. Z., Abdul Rashid, R., Ng, C. G., ... & Sulaiman, A. H. (2022, March). Prevalence of Anxiety and Depression among Psychiatric Healthcare Workers during the COVID-19 Pandemic: A Malaysian Perspective. In Healthcare (Vol. 10, No. 3, p. 532). MDPI.
Author Response
- Dear authors, Thank you for allowing me to review this important work. I have a few suggestions/queries. We thank the reviewer for his/her positive and constructive observations. Below, please find item-by-item responses to your comments, which are included verbatim.
- In the methodology section, the authors do not mention which databases the studies were extracted from. Please include this information. We specified all databases to which we have drown on studies, according the reviewer suggestion.
- The inclusion and exclusion criteria for the selected studies are not included. Inclusion/exclusion criteria have been already included in the text, as reported below: "The inclusion criteria were shown below: 1) RCTs or Observational studies; 2) professional health carers and patients who have contracted virus during a pandemic in the last 10 years; 3) Adults and elderly people with viral infection diagnosis; 4) suitable measures to assess intervention effectiveness on the clinical status, psychological and behavioural aspects. The exclusion criteria were the following: 1) participants with severe mental health symptoms such as psychosis or suicidality; 2) no English editing, and 3) no psychological interventions".
- There is a stark absence of a forest plot to summarize the findings of the meta-analysis. Please include one if possible. We included a forest plot (Figure 3), as reviewer suggested.
- The discussion section is rather simplistic and brief. It should be more scholarly in nature and discuss the novelty of the findings in a critical way. We rewrote the Discussion section, according to reviewer observations.
- There are many serious grammatical errors throughout the manuscript. The authors are advised to thoroughly proofread the manuscript before resubmission. According to reviewer suggestion, we obtained a review of the manuscript from a native English speaker.
- Important limitations and strengths of the meta-analysis must be mentioned in the manuscript. We reported limitations and strengths of the meta-analysis, as reviewer suggested.
- In the introduction section, lines 36-40, the authors posit that the pandemic has lead to many psychological consequences. The authors are guided to the following papers that also elaborate on the mental health outcomes of frontline workers during the pandemic:
- Francis B, Soon Ken C, Yit Han N, et al. Religious coping during the COVID-19 pandemic: gender, occupational and socio-economic perspectives among Malaysian frontline healthcare workers. Alpha Psychiatry. 2021;22(4):194-199.
- Narendra Kumar, M. K., Francis, B., Hashim, A. H., Zainal, N. Z., Abdul Rashid, R., Ng, C. G., ... & Sulaiman, A. H. (2022, March). Prevalence of Anxiety and Depression among Psychiatric Healthcare Workers during the COVID-19 Pandemic: A Malaysian Perspective. In Healthcare (Vol. 10, No. 3, p. 532). MDPI.
We added the aforesaid studies, as recommended by reviewer.
We thank the reviewer again for his/her suggestions that have certainly improved and enriched our work.
Round 2
Reviewer 2 Report
Thanks for opportunity to review revised manuscript entitled ' 'Psychological Interventions in a Pandemic Emergency: A Systematic Review and Meta-analysis of 10-year studies'' for JCM. I congratulate to authors. Author significantly improved manuscript from the previous submission. However, some revisions still required before publication of articles.
1. Authors must correct writing of partial eta-square (ηp2) along the manuscript.
2. Combine following first and second paragraph in Introduction to single paragraph ‘ ‘In the past 10 years’ viral epidemics have occurred, such as influenza caused by thevirus subtype H1N1 (2009), Middle East respiratory syndrome - MERS (2012), and Ebola virus disease (2014) [1, 2, 3]. At the end of 2019, a novel virus Severe Acute Respiratory Syndrome Coronavirus 2 33 (SARS-CoV-2) emerged around the world. Globally, response measures toke, as well as self-isolation, social distancing, and quarantines, augmenting further the distress
3. Combine following in a single paragraph ‘ ‘During SARS-CoV-2 pandemic, innovative technological know-how showed huge 61 strength to back up public health interventions [8]. 62 In light of these assumptions, providing an effective psychological assistance for pa-tients during a pandemic may impact positively on the mental health aspects. However, evidence-based advice on what would be effective in mitigating this impact are yet not clear. A rapid review conducted to detect the most effective interventions for reducing nurses’ psychological issues during outbreaks [9]; but the 14 article included were recom- mendations of organizational and individual self-care interventions, except a study which were a study describing a digital intervention and user satisfaction. Another rapid systematic review conducted to identify coping strategies among healthcare workers [10]; this study only included studies with at least healthcare workers, removing studies with smaller samples that might contribute to clinical practice and future research.’’
4. Move following information to under the search strategy ‘ ‘Prospero pre-registration run with n. CRD42020219191 (November 9, 2020), available at: https://www.crd.york.ac.uk/prospero/display_record.php?ID=CRD42020219191. Literature searching has been run by Scopus, MEDLINE (Web of Science), CINAHL Complete, PsycINFO and PubMed (from April 2020 to April 2021). The PRISMA guidelines for systematic review followed [11] and employed to evaluate quality of study. An inductive approach applied to the analysis [12].
5. Combine following in a single paragraph ‘ ‘The data synthesized according to PICO criteria [15], and they consisted in the testing for a relationship between the size and precision of study effects. The risk ratios for individual studies combined using a random effects meta-analysis. Co-authors supplied a complete summary and accurate organization of all studies extracting transparency, approach, methodology, and strengths/weaknesses [16, 17].’’
6. Remove following sentence from the results ‘ ‘According Statistical outcomes of each study reported in Table 2. Overall,’’ and begin with Post……..
Following sentence is not appropriate for discussion and must remove , because authors examined studies related to COVİD-19 pandemic ‘ ‘Authors According to work aim, evidences and effectiveness of psychological interventions applied during the last 10 years’ pandemics, including therapeutic outcomes and measurable psychological improvements reviewed.’’
7. Author must add limitations of their meta-analysis before conclusion section.
8. Last part of Discussion including ‘ ‘Closing, healthcare systems require a step-change in preparedness to deal with future pandemics. Psychological systems as well as medical systems of health care require huge investment and major upgrading in readiness for future pandemics. We found limited evidence for coping capacity. Predominantly, all the studies have a low risk of bias. However, that future studies will improve design quality to increase the trust in their outcomes.’’
9. Please also recheck the discussion carefully after focusing on COVİD-19 pandemic, it is still valid your interpretations. I did not observed much problem but checking it may be useful.
10. Please provide a proofreading document in the next submission from the MDPİ or other professional proofreading companies after above changes. I still observed a lot of English mistake along the manuscript. I personally recommend proofreadingpal (https://proofreadingpal.com/ You can check comments from here (https://www.trustpilot.com/review/proofreadingpal.com?utm_medium=trustbox&utm_source=MiniCarousel )

Author Response
- Thanks for opportunity to review revised manuscript entitled ' 'Psychological Interventions in a Pandemic Emergency: A Systematic Review and Meta-analysis of 10-year studies'' for JCM. I congratulate to authors. Author significantly improved manuscript from the previous submission. However, some revisions still required before publication of articles. We thank the reviewer for his/her positive and constructive observations. Below, please find item-by-item responses to your comments, which are included verbatim.
- Authors must correct writing of partial eta-square (ηp2) along the manuscript. Combine following first and second paragraph in Introduction to single paragraph ‘ ‘In the past 10 years’ viral epidemics have occurred, such as influenza caused by thevirus subtype H1N1 (2009), Middle East respiratory syndrome - MERS (2012), and Ebola virus disease (2014) [1, 2, 3]. At the end of 2019, a novel virus Severe Acute Respiratory Syndrome Coronavirus 2 33 (SARS-CoV-2) emerged around the world. Globally, response measures toke, as well as self-isolation, social distancing, and quarantines, augmenting further the distress. According to reviewer suggestions, we made the aforesaid improvements.
- Combine following in a single paragraph ‘ ‘During SARS-CoV-2 pandemic, innovative technological know-how showed huge 61 strength to back up public health interventions [8]. 62 In light of these assumptions, providing an effective psychological assistance for pa-tients during a pandemic may impact positively on the mental health aspects. However, evidence-based advice on what would be effective in mitigating this impact are yet not clear. A rapid review conducted to detect the most effective interventions for reducing nurses’ psychological issues during outbreaks [9]; but the 14 article included were recom- mendations of organizational and individual self-care interventions, except a study which were a study describing a digital intervention and user satisfaction. Another rapid systematic review conducted to identify coping strategies among healthcare workers [10]; this study only included studies with at least healthcare workers, removing studies with smaller samples that might contribute to clinical practice and future research.’’ We made the aforesaid improvement, as reviewer recommended.
- Move following information to under the search strategy ‘ ‘Prospero pre-registration run with n. CRD42020219191 (November 9, 2020), available at: https://www.crd.york.ac.uk/prospero/display_record.php?ID=CRD42020219191. Literature searching has been run by Scopus, MEDLINE (Web of Science), CINAHL Complete, PsycINFO and PubMed (from April 2020 to April 2021). The PRISMA guidelines for systematic review followed [11] and employed to evaluate quality of study. An inductive approach applied to the analysis [12]. We moved the denoted information under the Search strategy subsection.
- Combine following in a single paragraph ‘ ‘The data synthesized according to PICO criteria [15], and they consisted in the testing for a relationship between the size and precision of study effects. The risk ratios for individual studies combined using a random effects meta-analysis. Co-authors supplied a complete summary and accurate organization of all studies extracting transparency, approach, methodology, and strengths/weaknesses [16, 17].’’ We combined the aforesaid sentences in a single paragraph, as reviewer suggested.
- Remove following sentence from the results ‘ ‘According Statistical outcomes of each study reported in Table 2. Overall,’’ and begin with Post…….. We deleted the aforesaid sentence.
- Following sentence is not appropriate for discussion and must remove , because authors examined studies related to COVİD-19 pandemic ‘ ‘Authors According to work aim, evidences and effectiveness of psychological interventions applied during the last 10 years’ pandemics, including therapeutic outcomes and measurable psychological improvements reviewed.’’ The aforesaid sentence has been correct.
- Author must add limitations of their meta-analysis before conclusion section. We clarified the limitations of the meta-analysis, as reviewer suggested.
- Last part of Discussion including ‘ ‘Closing, healthcare systems require a step-change in preparedness to deal with future pandemics. Psychological systems as well as medical systems of health care require huge investment and major upgrading in readiness for future pandemics. We found limited evidence for coping capacity. Predominantly, all the studies have a low risk of bias. However, that future studies will improve design quality to increase the trust in their outcomes.’’ We improved these sentences, as reviewer observed.
- Please also recheck the discussion carefully after focusing on COVİD-19 pandemic, it is still valid your interpretations. I did not observed much problem but checking it may be useful. We re-check the Discussion section, according to reviewer recommendation.
- Please provide a proofreading document in the next submission from the MDPİ or other professional proofreading companies after above changes. I still observed a lot of English mistake along the manuscript. I personally recommend proofreading pal (https://proofreadingpal.com/ You can check comments from here (https://www.trustpilot.com/review/proofreadingpal.com?utm_medium=trustbox&utm_source=MiniCarousel ). According to reviewer suggestion, we re-obtained a review of the manuscript from a native English speaker. We thank the reviewer again for his/her suggestions that have certainly improved and enriched our work.
Reviewer 3 Report
Dear authors,
1) The forest plot included does not include important information such as a heterogeneity measure and summary diamond. It is also not properly labelled, making it difficult to interpret.
2) There are still several major grammatical errors in the article. Please proofread again.
3) The various grammatical and constructional errors in the discussion section makes it a challenge to fully comprehend what the authors are trying to deliver. The authors are encouraged to write the discussion with more clarity.
Author Response
- Dear authors, The forest plot included does not include important information such as a heterogeneity measure and summary diamond. It is also not properly labelled, making it difficult to interpret. We thank the reviewer for his/her positive and constructive observations. Below, please find item-by-item responses to your comments, which are included verbatim. The forest plot has been revised in order to clarify the psychological intervention effect of the included studies.
- There are still several major grammatical errors in the article. Please proofread again. According to reviewer suggestion, we re-obtained a review of the manuscript from a native English speaker.
- The various grammatical and constructional errors in the discussion section makes it a challenge to fully comprehend what the authors are trying to deliver. The authors are encouraged to write the discussion with more clarity. We improved the Discussion section, as reviewer suggested. We thank the reviewer again for his/her suggestions that have certainly improved and enriched our work.